# Action of Multiple Rice β-Glucosidases on Abscisic Acid Glucose Ester

**DOI:** 10.3390/ijms22147593

**Published:** 2021-07-15

**Authors:** Manatchanok Kongdin, Bancha Mahong, Sang-Kyu Lee, Su-Hyeon Shim, Jong-Seong Jeon, James R. Ketudat Cairns

**Affiliations:** 1School of Chemistry, Institute of Science, Center for Biomolecular Structure, Function and Application, Suranaree University of Technology, Nakhon Ratchasima 30000, Thailand; D5610464@g.sut.ac.th; 2Graduate School of Biotechnology and Crop Biotech Institute, Kyung Hee University, Yongin 17104, Korea; berker0213@yahoo.com (B.M.); kyuri93@khu.ac.kr (S.-K.L.); hiyaho15@khu.ac.kr (S.-H.S.)

**Keywords:** abscisic acid, β-Glucosidase, glycosylation, *Oryza sativa*, phytohormone conjugates

## Abstract

Conjugation of phytohormones with glucose is a means of modulating their activities, which can be rapidly reversed by the action of β-glucosidases. Evaluation of previously characterized recombinant rice β-glucosidases found that nearly all could hydrolyze abscisic acid glucose ester (ABA-GE). Os4BGlu12 and Os4BGlu13, which are known to act on other phytohormones, had the highest activity. We expressed *Os4BGlu12*, *Os4BGlu13* and other members of a highly similar rice chromosome 4 gene cluster (*Os4BGlu9*, *Os4BGlu10* and *Os4BGlu11*) in transgenic Arabidopsis. Extracts of transgenic lines expressing each of the five genes had higher β-glucosidase activities on ABA-GE and gibberellin A_4_ glucose ester (GA_4_-GE). The β-glucosidase expression lines exhibited longer root and shoot lengths than control plants in response to salt and drought stress. Fusions of each of these proteins with green fluorescent protein localized near the plasma membrane and in the apoplast in tobacco leaf epithelial cells. The action of these extracellular β-glucosidases on multiple phytohormones suggests they may modulate the interactions between these phytohormones.

## 1. Introduction

β-Glucosidases (EC. 3.2.1.21, β-D-glucopyranosidases) are enzymes that hydrolyze glycosidic linkages to release glucose from the non-reducing termini of oligosaccharides and aryl and alkyl glucosides [1]. β-Glucosidases have been found in wide range of living organisms, from bacteria and archaea to multicellular eukaryotes, including mammals and plants. A number of crucial biological reactions in living cell are catalyzed by these enzymes, particularly in plants [2].

Among β-glucosidase enzymatic abilities, hydrolysis of plant phytohormone glyco- conjugates was addressed in this study. Abscisic acid (ABA) is a crucial plant phytohormone playing a role in biological processes, especially responses to adverse stresses, such as drought, salinity, cold and pathogen attack [3]. In planta, the ABA level is regulated by biosynthesis and catabolism [4]. In the de novo biosynthesis of ABA, zeaxanthin is generated at plastids and consequently changed to xanthoxin. It then translocates from the plastids to the cytoplasm and is converted to ABA. The translocation of ABA between cells, tissues and organs also plays an important role in whole plant physiological response to stress conditions. ABA is a weak acid, which can diffuse passively across biological membranes when it is protonated [5], but membrane transporters have been identified for transport between the neutral cytoplasm and the acidic apoplast [4]. Since accumulation of de novo synthesized ABA negatively affects the biological function of plant cells, ABA is metabolized to control its level.

One process that controls the activity and localization of ABA is glycosylation to form ABA glucose ester (ABA-GE) [4,6]. In Arabidopsis, ABA can be converted to ABA-GE by UDP-glucosyltransferase 71C5 to allow storage in membranous organelles, including the endoplasmic reticulum and vacuoles [7,8,9]. To increase the level of ABA in response to abiotic stress, such as cold, water-deficiency and salt, ABA-GE can be hydrolyzed to release free ABA, which can pass through membranes from acidic compartments, like the apoplast and vacuole, to enter the cytoplasm [10]. β-Glucosidase was reported to release ABA from the physiologically inactive ABA-GE pool in the leaf apoplast [11]. An Arabidopsis β-glucosidase (AtBG1) was found to hydrolyze ABA-GE to release free ABA [12]. Loss of AtBG1 affected stomata closure, and resulted in early germination and sensitivity to abiotic stress. AtBG1 was localized to the ER, and ABA immunoreactivity was reported to accumulate in the cytoplasm near ER tubules [13]. Heterologous expression of AtBG1 in creeping bent grass increased ABA levels and enhanced drought tolerance compared with wild type [14]. A second Arabidopsis β-glucosidase (AtBG2), which is localized in the vacuole, was also found to hydrolyze ABA-GE to produce free ABA during dehydration stress [10].

The rice genome contains over 30 glycoside hydrolase family 1 (GH1) genes encoding β-glucosidases and their homologues, including β-mannosidases and transglucosidases [2,15]. The rice GH1 enzymes were given names based on their gene location with a chromosome number (Os1-12) and a running number starting from the top of chromosome 1 to the end of chromosome 12 (BGlu1-38) [15]. They were divided into eight phylogenetic clusters which contained both Arabidopsis and rice proteins, At/Os1 to 8, based on protein sequence similarity (Figure 1A). Initial studies indicated that several of these enzymes have activity on cell-wall-derived oligosaccharides, as well as assorted glucosides, and a few were reported to show significant transglycosylation activity [15,16,17]. Later, Os9BGlu31 was found to have much higher ability to transfer glucose between substrates, including acidic phytohormones, than to hydrolyze glucosides [18]. Among rice β-glucosidases that act on phytohormones, Os4BGlu13 was found to hydrolyze tuberonic acid (TA) glucoside (TAG) to release active TA, leading to its designation as *Oryza sativa* TAG β-glucosidase 1 (OsTAGG1) [19]. The closely related isoenzyme Os4BGlu12 was also found to hydrolyze TAG and was designated TAGG2, but was later found to hydrolyze salicylic acid (SA) glucoside (SAG) more efficiently [20,21]. Hua et al. (2015) reported that Os4BGlu13 also hydrolyzes gibberellin A4 glucose ester (GA_4_-GE), in addition to SAG and TAG [22].

Although it is known that some enzymes can hydrolyze ABA-GE in plants like Arabidopsis [10,12], it is not clear how many enzymes in one plant are capable of this function. To assess which rice β-glucosidases may act on ABA-GE, we screened several GH1 enzymes recombinantly expressed in *Escherichia coli* for ABA-GE hydrolysis. Although most of these enzymes hydrolyze ABA-GE, we found that Os4BGlu12 and Os4BGlu13 hydrolyzed ABA-GE better than the other rice β-glucosidases tested. A set of closely related rice β-glucosidases, belonging to GH1 phylogenetic cluster At/Os7, were found to hydrolyze ABA-GE via exogenous expression in Arabidopsis and their green fluorescent protein (GFP)-tagged proteins were localized near the plasma membrane and cell wall when expressed in tobacco leaves. Expression of these enzymes also modulated the response of the transgenic Arabidopsis to salt and drought stress.

## 2. Results

### 2.1. Hydrolysis of ABA-GE by Rice Enzymes Belonging to GH1 Family

To identify which enzymes may hydrolyze ABA-GE, several GH1 β-glucosidases expressed in *E. coli* were screened. Rice Os4BGlu12, Os4BGlu13, Os1BGlu4, Os3BGlu7, Os4BGlu18, Os3BGlu6, Os7BGlu26, Os9BGlu31 and barley βII showed activity with ABA-GE, albeit low in some cases, as shown in Table 1. Os4BGlu12 and Os4BGlu13, which have previously been shown to hydrolyze other phytohormone gluco-conjugates [19,20,21,22], showed the highest hydrolysis activity with ABA-GE. It was noted that Os4BGlu12 and Os4BGlu13 belong to GH1 phylogenetic cluster At/Os7, which also contains Os4BGlu9, Os4BGlu10 and Os4BGlu11, the genes for which are found within 60 kb of those for Os4BGlu12 and Os4BGlu13 on rice chromosome 4, suggesting a repetitive gene duplication (Figure 1) and genes that might have redundant or similar functions. However, attempts to express Os4BGlu9, Os4BGlu10 and Os4BGlu11 in *E. coli* failed to produce active enzymes (data not shown), so their activities could not be explored in the same manner.

### 2.2. Kinetic Analysis of Os4BGlu12 and Os4BGlu13 Hydrolysis of ABA-GE

Kinetic parameters for hydrolysis of ABA-GE were determined for Os4BGlu12 and Os4BGlu13 (Table 2). Os4BGlu13 has higher catalytic efficiency (*k*_cat_/*K*_M_ = 12.4 mM^−1^s^−1^) for hydrolysis of ABA-GE than Os4BGlu12 (*k*_cat_/*K*_M_ = 0.689 mM^−1^s^−1^). This reflects the 6-fold lower *K*_M_ and 2.7-fold higher *k*_cat_ of Os4BGlu13 compared to Os4BGlu12.

### 2.3. Subcellular Localization Examination of Rice β-Glucosidases Os4BGlu9, Os4BGlu10, Os4BGlu11, Os4BGlu12 and Os4BGlu13

*Nicotiana benthamiana* leaves co-infiltrated with expression vectors producing rice β-glucosidase-GFP fusions and red fluorescent protein (RFP)-Korrigan exhibited partially overlapping signals between the GFP and RFP signals (Figure 2). The red Kor signal is expected to be anchored to the inner surface of the plasma membrane, thereby marking the plasma membrane position. Most of the green signals appeared to colocalize with the plasma membrane and cell wall. This fluorescence microscopy of the fusion proteins supports the localization of Os4BGlu9, Os4BGlu10, Os4BGlu11, Os4BGlu12 and Os4BGlu13 in the apoplast near the plasma membrane and cell wall, although some may have been localized to intracellular organelles, such as endoplasmic reticulum, that was pressed tightly against the cell membrane.

To clarify the localization further, we established rice lines expressing Os4BGlu10 and Os4BGlu13 with GFP fused to their C-termini. The Os4BGlu10-GFP and Os4BGlu13-GFP green signals colocalized with the plasma membrane marker FM4-64 (Appendix A). Upon plasmolysis to separate the plasma membrane from the cell wall, the bulk of the green fluorescent signal colocalized with the membrane, but a significant portion was left in the cell wall (Appendix A), suggesting partitioning between the membrane and cell wall.

### 2.4. Generation of Transgenic Arabidopsis Plants Expressing Rice β-Glucosidases

The pH7FWG2 expression vectors containing the Os4BGlu9, Os4BGlu10, Os4BGlu11, Os4BGlu12 or Os4BGlu13 cDNA under control of the CaMV35S promoter (Figure 3A) were transformed into *Arabidopsis thaliana* Col-0 and selected to obtain homozygous transgenic Arabidopsis lines expressing rice β-glucosidases. RT-PCR (Figure 3B) gave the expected band amplified for each specific rice gene in the respective expression line, and no band for rice β-glucosidase expression was detected in wild-type lines.

### 2.5. In Vitro Hydrolysis of pNPGlc, ABA-GE and GA_4_-GE by Extracts of Transgenic Arabidopsis Expressing Rice β-Glucosidases

The extracts of the two independent lines of transgenic Arabidopsis expressing rice β-glucosidases Os4BGlu9, Os4BGlu10, Os4BGlu11, Os4BGlu12 and Os4BGlu13 had higher hydrolysis activity with the synthetic substrate *p*-nitrophenyl- β-D-glucopyranoside (*p*NPGlc) than the wild type plant extracts, although for the Os4BGlu9 lines, the difference was not significant (Figure 4A). Among the extracts of plants expressing rice β-glucosidases, the lines of transgenic Arabidopsis expressing Os4BGlu10-13 showed almost two-fold the activity of wild type on *p*NPGlc, while extracts of the lines expressing Os4BGlu9 had 10–40% higher activity than those of the control plants.

Since the Os4BGlu12 and Os4BGlu13 expressed in *E. coli* had high activity on ABA-GE and Os4BGlu13 was previously shown to have high activity on GA_4_-GE [22], we tested the extracts activities on these two phytohormone glucose-conjugates (Figure 4B,C). The extracts of the two independent lines of transgenic Arabidopsis expressing rice β-glucosidases Os4BGlu12 and 13 had the highest hydrolysis activities for ABA-GE and GA_4_-GE (approximately 2.3–4 times that of control plants), while extracts from plants expressing Os4BGlu9-11 had 1.5–3 times the activity of controls. Clearly, expression of each gene increased the phytohormone glucose ester β-glucosidase activity in the plant.

### 2.6. Growth of Arabidopsis Seedlings Expressing Rice β-Glucosidases upon Stress Treatments

Since ABA is known to be crucial in salt stress and drought response [3,4,5], we tested the response to NaCl and PEG stress of Arabidopsis lines expressing rice β-glucosidases. Under unstressed conditions, the lines expressing rice β-glucosidases were indistinguishable from wild type plants (Appendix A). Salt (125 mM NaCl) and drought (20% PEG8000) stress shortened the roots and shoots in wild type plants, but this effect was significantly smaller for the transgenic plants heterologously expressing the rice β-glucosidases (Figure 5).

## 3. Discussion

Based on protein sequence similarity, Os4BGlu9, Os4BGlu10, Os4BGlu11, Os4BGlu12 and Os4BGlu13 are closely related and fall into a rice chromosome 4-specific subclade of the protein-sequence-based phylogenetic cluster At/Os7 (Figure 1) [15]. Among these, Os4BGlu12 and Os4BGlu13 are two of the rice β-glucosidases that have been produced by recombinant expression in *E. coli*. Both of these enzymes have been isolated from plants based on their hydrolysis of tuberonic acid glucoside and have been shown to hydrolyze the glucoside of the phytohormone SA, in addition to cell wall derived oligosaccharides [17,19,20,21]. Os4BGlu13 was also isolated from rice plants based on its ability to hydrolyze GA_4_-GE [22].

Although two β-glucosidases, AtBG1 and AtBG2, have been identified to act on ABA-GE in Arabidopsis [10,12], no clear orthologues of these enzymes have been identified in rice. The rice β-glucosidases Os1BGlu4, Os3BGlu7, Os4BGlu12, Os4BGlu13, Os4BGlu18 and Os7BGlu26 that were expressed in *E. coli* had significant activity to release glucose from ABA-GE, with Os4BGlu13 and Os4BGlu12 exhibiting highest hydrolysis activity. Although Os9BGlu31 has little activity to release glucose, it is known to transglycosylate acidic phytohormones [18], allowing us to use it to produce ABA-GE. This result suggests several β-glucosidase isoenzymes may affect ABA-GE metabolism in the plant, although knockouts of single genes have been shown to have significant effects on ABA metabolism and stress response [10,12]. Os7BGlu26 β-mannosidase had >10-fold lower activity than most of the other GH1 hydrolases tested, indicating some GH1 hydrolases have relatively little activity against ABA-GE, despite its wide acceptability as a substrate.

Recently, Os3BGlu6 expression was shown to modulate ABA-GE levels in the plant and affect ABA-related plant traits, and its overexpression increased ABA-GE hydrolysis activity in rice leaf extracts [23], although it had relatively low level of activity on ABA-GE in our assay. The dwarf phenotype of the Os3BGlu6 knockout line in that report suggests that it may increase plant growth via release of GA from GA_4_-GE, upon which it has high activity [24]. The relatively high levels of activity of Os4BGlu12 and Os4BGlu13 supplement the previous investigations showing their hydrolysis activity toward phytohormone gluco-conjugates. Recently, knockout mutation of Os4BGlu10 provided evidence for its role in ABA-GE metabolism, as well [25]. Thus, we hypothesized a role for rice GH1 phylogenetic cluster At/Os7 enzymes as phytohormone glucoconjugate β-glucosidases, particularly those in the closely related subclade of genes on rice chromosome 4 (Os4BGlu9-13).

Kinetic characterization of hydrolysis of ABA-GE indicated Os4BGlu13 had an 18-fold higher *k*_cat_/*K*_M_ value with ABA-GE than Os4BGlu12. Since the levels of ABA-GE in the plant are expected to remain much below the *K*_M_ values of these enzymes, these *k*_cat_/*K*_M_ values would reflect the relative rates of hydrolysis in the plant, suggesting that, in principle, expression of Os4BGlu13 could have a greater effect on ABA-GE levels than Os4BGlu12. The *k*_cat_/*K*_M_ of Os4BGlu13 for ABA-GE of 12.4 mM^−1^s^−1^ is higher than that for TAG (6.68 mM^−1^s^−1^), GA_4_-GE (3.63 mM^−1^s^−1^) and SAG (0.88 mM^−1^s^−1^) [23], indicating higher specificity for ABA-GE. Nonetheless, the similar values for these different phytohormones suggest that Os4BGlu13 could act on multiple phytohormone conjugates in the plant, including ABA-GE. The expression of Os4BGlu9-13 in Arabidopsis also increased ABA-GE and GA_4_-GE hydrolysis more than that of the general synthetic substrate *p*-nitrophenyl-β-d-glucoside, suggesting they are selective for the phytohormone glucose esters.

Among the proteins encoded in the rice genome, the sequence of Os4BGlu12 was the most similar to that of a rice cell wall β-glucosidase determined by protein sequencing, suggesting it is localized to the cell wall [15,25]. The fluorescence observed from Os4BGlu9-GFP, Os4BGlu10-GFP, Os4BGlu11-GFP, Os4BGlu12-GFP and Os4BGlu13-GFP relative to RFP-KOR1 in *N. benthamiana* leaf epithelial cells and also from Os4BGlu10-GFP and Os4BGlu13-GFP in coleoptile cells of transgenic rice by confocal microscopy supports the localization of these β-glucosidases in the apoplast between the plasma membrane and cell wall, although some may also be localized in organelles close to the plasma membrane. Overexpression of the secreted β-glucosidases in the plant may increase the enzyme level throughout the secretory pathway, from the endoplasmic reticulum (ER) to the Golgi and apoplast. The fact that Os4BGlu12 and Os4BGlu13 are produced in active form in *E. coli* without eukaryotic posttranslational modification, suggests the enzymes could be active upon folding in the ER. Os4BGlu12 and Os4BGlu13 have pH optima around pH 5 and low activity at the neutral pH of the ER and Golgi [15,19,24], so they may have a relatively small effect on β-glucosidase activity in these compartments. Nevertheless, over production of secreted β-glucosidase might affect pools of ABA-GE sequestered in the ER [4,7], as well as those in the apoplast, if the enzyme concentration is increased in the intracellular compartment.

At first, we had expected transgenic Arabidopsis seedlings expressing Os4BGlu9-13 might have shorter shoots and stems due to release of ABA from ABA-GE [26]. However, ABA can have positive effects on root and shoot lengths at low concentration but negative effects at high concentration [27,28]. It is worth noting that GH1 enzymes have been shown to catalyze transglycosylation as well as hydrolysis [18], suggesting that in some conditions these enzymes may catalyze a net glycosylation of the phytohormone, rather than deglycosylation.

ABA is known to mediate the response to abiotic stresses, including the NaCl response [29]. The rice β-glucosidase expression lines had root and shoot lengths longer than control plants after growth in 125 mM NaCl (Figure 5A,C), supporting the idea that rice β-glucosidases in family GH1 phylogenetic cluster At/Os7 are involved in stress responses. In the case of drought stress, simulated by PEG osmotic stress, roots and shoots of the β-glucosidase expression lines were significantly longer than those of wild type, as well (Figure 5B). An Os4BGlu10 knockout rice line characterized by Ren and colleagues [30] showed decreased tolerance to salt and drought stress, consistent with the expression of Os4BGlu10 and its homologues in Arabidopsis improving salt stress resistance.

ABA and gibberellins (GAs) are generally thought to have antagonistic roles in plant development and response to stress [31]. For instance, a gain of function mutant of the Gibberellic Acid Insensitive (GAI) DELLA protein, a negative regulator of GA signaling, was shown to cause improved resistance to drought, by decreasing stomata aperture in the presence of ABA [32]. It apparently interacts with the ABA signaling positive regulator ABF2 (ABA-responsive element binding transcription factor 2), revealing the crosstalk between ABA and GA in response to drought tolerance. This finding hints at a fine-tuned balance between signaling of different phytohormones in response to environmental stress. The presence of β-glucosidases that can activate the glucosylated storage forms of multiple phytohormones may provide a way to buffer the effects of these pathways to help maintain that balance.

In summary, several rice β-glucosidases have the ability to hydrolyze ABA-GE, some of which are known to act on other phytohormone gluco-conjugates. Rice family GH1 phylogenetic cluster At/Os7 members Os4BGlu9, Os4BGlu10, Os4BGlu11, Os4BGlu12 and Os4BGlu13 are β-glucosidases located around the cell wall. Based on their activities in plant extracts, these enzymes may hydrolyze ABA-GE and GA_4_-GE in the apoplast to release free phytohormones that may then enter into the cell. The action of these enzymes on multiple phytohormones may help fine-tune the response to stress and phytohormones and suggests intricate regulation of phytohormone activity by the enzymes involved in glycosylation and deglycosylation.

## 4. Materials and Methods

### 4.1. Production of ABA-GE by Os9BGlu31 Transglucosidase

Os9BGlu31 transglycosidase was produced as previously described [18] and used to transfer glucose from *p*NPGlc to ABA to generate ABA-GE (Patent pending Thai patent application 1801003832). Briefly, 10 µg of Os9BGlu31 enzyme catalyzed the reaction of 10 mM ABA with 10 mM *p*NPGlc in citrate buffer, pH 4.5, at 37 °C overnight. The reaction was stopped by boiling for 5 min. The ABA-GE product was purified by silica gel chromatography in 8% methanol 2% acetic acid and C18 reverse phase chromatography with increasing methanol in water. The identity and purity of the ABA-GE was verified by Nuclear Magnetic Resonance (NMR) and Ultra-High Performance Liquid Chromatography (UHPLC).

### 4.2. The Hydrolysis Activity of Rice GH 1 Enzymes toward ABA-GE

The rice enzymes belonging to GH1, including Os1BGlu4 [33], Os3BGlu6 [34], Os3BGlu7 [19], Os4BGlu12 [15,18], Os4BGlu13 [22], Os4BGlu18 [35], Os7BGlu26 [36], Os9BGlu31 [18] and barley β2 [36] were expressed and purified by immobilized metal affinity chromatography (IMAC), as previously described. The purified enzymes were tested for ABA-GE hydrolysis in reactions containing 0.5 µg of enzyme with 1 mM ABA-GE in 50 mM sodium acetate, pH 5.0, incubated at 30 °C for 30 min. The reactions were stopped by boiling for 5 min, and the glucose released was quantified by the peroxidase/glucose oxidase assay method (PGO assay, Sigma Aldrich, St. Louis, MO, USA) measuring absorbance at 405 nm and comparing to a glucose standard curve. The blank was a control reaction incubated without enzyme and processed in the same way.

### 4.3. Kinetic Study of Os4BGlu12 and Os4BGlu13 with ABA-GE

Os4BGlu12 and Os4BGlu13 were purified by IMAC, followed by cleavage of the N-terminal thioredoxin and His_6_ tags with enterokinase [22]. The fusion tags were removed by IMAC to yield >90% pure proteins (Appendix A). All of kinetic parameters were determined in triplicate reactions. The assays were done at 30 °C in 50 mM sodium acetate (NaOAc), pH 5. The glucose released was determined as described in the previous section. A time course was conducted to ensure that initial rates were measured. Kinetic parameters (*K*_M_ and *V*_max_) were calculated by fitting the rate of product formation and substrate concentrations by nonlinear regression of the Michaelis–Menten curves with GraFit 5.0. The apparent *k*_cat_ values were calculated by dividing the *V*_max_ by the total amount of enzyme in the reaction.

### 4.4. Subcellular Localization of Os4BGlu-GFP Fusion Proteins in Plant Cells

To determine subcellular localization of rice Os4BGlu9, Os4BGlu10, Os4BGlu11, Os4BGlu12 and Os4BGlu13, the entire open reading frames of their cDNA sequences not including the stop codons were amplified by polymerase chain reaction (PCR) with a proofreading EF-Taq polymerase (SolGent, Daejeon, Korea) and the primers listed in Appendix A. The respective PCR products were further cloned into the pENTR/D-TOPO^®^ vector by the supplier’s protocol (Invitrogen, Thermo Fischer Scientific, Waltham, MA, USA). Afterward, those amplicons were cloned into the binary vector pH7FWG2 [37] to generate recombinant vectors encoding chimeric proteins that fused the β-glucosidases’ C-termini to GFP with their expression controlled by a CaMV35S promoter. In addition, a vector encoding RFP fused to Korrigan1 (GenBank: AK318891) was used to provide a plasma membrane protein marker [38]. All constructs were introduced into *N. benthamiana* leaves together with a Tomato Bushy Stunt Virus P19 protein expression vector by an *Agrobacterium*-mediated infiltration method [39]. Subcellular localization was monitored at day 3 to 5 after infiltration under a scanning confocal microscope (LSM 510 META; Carl Zeiss, Jena, Germany). To monitor GFP, the argon laser was used for excitation at 488 nm wavelength and GFP filter for emission at 515–530 nm; to monitor RFP and FM4-64: He-Ne laser for excitation at 543 nm and mChFP filters for emission at 580–700 nm.

For further examination of subcellular localization of the members of the cluster At/Os7 of GH1 in rice, we cloned Os4BGlu10 and Os4BGlu13 into the pENTR/D-TOPO^®^ vector, then subcloned in-frame between the maize (*Zea mays*) Ubiquitin1 promoter and GFP in the pIPKb002 binary vector, respectively [40]. The final constructs were transformed into rice callus via *Agrobacterium*-mediated transformation [41]. Coleoptile samples from their progeny seedlings 5 days after germination on 1/2 Murashige and Skoog media containing hygromycin were observed with a confocal microscope (LSM 510 META; Carl Zeiss GmbH, Jena, Germany). Coleoptile tissues were stained with a plasma membrane marker FM4-64 (Invitrogen) by the manufacturer’s instructions. Thereafter, 20% sucrose was added on one side of the coverslip for 5–10 min for plasmolysis.

### 4.5. Construction of Plant Expression Vectors and Arabidopsis Transformation

Transgenic Arabidopsis plants expressing the 5 rice β-glucosidase proteins were generated, as described for Os4BGlu14, Os4BGlu16 and Os4BGlu18 by Baiya et al. [39]. Briefly, the full-length cDNAs were amplified with a *Pfu* DNA polymerase (SolGent) from japonica rice (cv. Nipponbare) cDNA clones (J013092D04 for *Os4BGlu9*, accession number AK066908; J013041M21 for *Os4BGlu10*, accession number AK065793; J090089B01 for *Os4BGlu11*, accession number AK242955; J023122G03 for *Os4BGlu12*, accession number AK100820; and J023066D17 for *Os4BGlu13* accession number AK070962), which were ordered from the Knowledge-based *Oryza* Molecular biological Encyclopedia (KOME) [42]. The primer pairs used for PCR amplification are given in Appendix A. The respective PCR products were cloned into the pENTR/D-TOPO vector and the inserts were verified by sequencing. The cDNA inserts were cloned via LR clonase recombination (Invitrogen, Thermo Fischer Scientific) into the Gateway binary vector pGWB502, containing CaMV35S promoter. All recombinant binary vectors for expression were introduced into *A. tumefaciens* strain GV3101 by electroporation and subsequently transformed into Arabidopsis plant ecotype Columbia (Col-0) by the floral dip method, as described by Clough and Bent [43]. The T1 transformants were selected on plates containing 25 mg/L hygromycin. The homozygous transgenic plants were selected by choosing T3 transformants from T2 parents from which >99% of seeds germinated on hygromycin.

### 4.6. RNA Isolation and RT-PCR Analysis

Gene expression levels were validated in 1-month-old leaves of the two independent transgenic Arabidopsis lines expressing one of the 5 β-glucosidase genes of the At/Os7 phylogenetic cluster. Three to four leaves were detached from a single plant and ground in liquid nitrogen. Total RNA was extracted from 10 mg of the powdered leaf sample in TRIzol Reagent, as described in the company protocol (Thermo Fischer Scientific). The RNA concentration was determined from 260 nm absorbance and 1 μg was used for reverse transcription. The isolated RNA extracts were reverse-transcribed in 20 μL reactions with an oligo-dT primer and a First Strand cDNA Synthesis Kit (Roche, Mannheim, Germany). The first-strand cDNAs (1 μL) were used as templates in the PCR with the primers listed in Appendix A. PCR reactions using *Taq* polymerase were conducted for 28–35 cycles, depending on the gene expression level. The Arabidopsis ubiquitin gene-specific primers were used as the internal control [44]. PCR products were visualized by agarose gel electrophoresis with ethidium bromide staining.

### 4.7. Growth and Treatments of Transgenic Arabidopsis

Two independent lines of transgenic Arabidopsis expressing rice β-glucosidases Os4BGlu9, Os4BGlu10, Os4BGlu11, Os4BGlu12, Os4BGlu13 and control plants were grown. Seeds were surface sterilized with 80% ethanol for 20 min followed by 10% Clorox, and washed with water 5 times. They were grown on ½ MS (HiMedia, Mumbai, India) containing 1% (*w*/*v*) sucrose and 1% (*w*/*v*) phytagel (HiMedia). The lines were grown at 23 °C in a culture room 16-h/8/h light/dark cycle under fluorescent lamps.

To test whether heterologous expression of the rice β-glucosidases affects NaCl and osmotic stress responses, plants were grown in ½ MS medium for 7 days and transplanted to ½ MS containing 125 mM NaCl or 20% PEG (molecular weight 8000; Sigma-Aldrich, St Louis, MO, USA). Water potential was lowered by pouring approximately 20 mL of PEG solution on top of an equal volume of solidified nutrient agar in a Petri plate, according to a previously described protocol [45]. MES buffer was added to stabilize the pH of the media. Root and shoot growth were measured 5 days after the plants were transplanted. To quantify root and shoot lengths at the end of each treatment, three independent sets of 20 plants each were measured for each line.

### 4.8. Extraction of Total Protein from Transgenic Arabidopsis and Determination of β-Glucosidase Activity

Two independent lines of transgenic Arabidopsis expressing rice β-glucosidases Os4BGlu9, Os4BGlu10, Os4BGlu11, Os4BGlu12, Os4BGlu13 and control plants were grown on ½ MS plates at 23 °C in a culture room with 80% relative humidity and a 16-h/8-h light/dark cycle for 7 days. Three separate extractions were made per line. Fresh Arabidopsis seedlings (10 mg) were ground in liquid nitrogen and mixed with 1 mL of lysis buffer (150 mM NaCl in 20 mM Tris-HCl, pH 8.0, containing 1 mM phenylmethylsulfonyl fluoride) by vortexing for 5 min, after which they were sonicated with ultrasonic output 40 W and ultrasonic frequency 35 kHz (Ultrasonic Bath, DT series, Bandelin, Berlin, Germany) on ice for 30 min. The suspensions were centrifuged at 20,400× *g* 10 min, and the supernatant was analyzed for protein concentration and hydrolysis activities toward *p*NPGlc, ABA-GE and GA_4_-GE. The protein concentrations were assayed by the Bradford method [41] with bovine serum albumin (BSA) as the standard.

Protein extracts (10 µg of total protein) in lysis buffer were incubated with 1 mM *p*NPGlc in 50 mM NaOAc, pH 5, at 30 °C for 8 h. The reactions were stopped by adding 2 M sodium carbonate (Na_2_CO_3_). The released *p*-nitrophenol (*p*NP) was quantified by measuring the absorbance at 405 nm (A_405_) with a microplate reader (Thermo Labsystems, Helsinki, Finland) and comparing to a pNP standard curve.

The hydrolysis of ABA-GE and GA_4_-GE was determined by similar reactions in 50 mM NaOAc buffer, pH 5, which were incubated for 4 h at 30 °C, then boiled 5 min, and the glucose was determined with a peroxidase/glucose oxidase-based glucose assay (PGO assay), as described in Section 4.2.

### 4.9. Statistical Analysis

Statistical analysis was performed using GraphPad Prism software (GPW7). Data are plotted as mean ± standard deviation of three biological replicates. To verify the significance of differences between the Arabidopsis expressing the rice β-glucosidases with control wild type lines, significance was evaluated via the SPSS statistics software with one-way ANOVA followed by post hoc Scheffe’s test method at a significance level of *p* < 0.05.

## Figures and Tables

**Figure 1 ijms-22-07593-f001:**
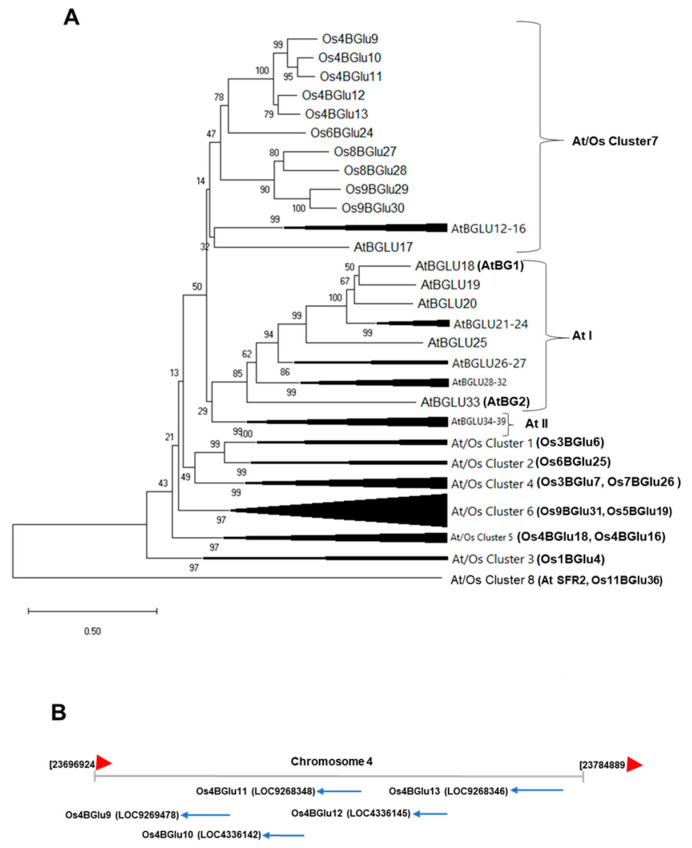
Phylogenetic and genomic relationships between rice family GH1 cluster At/Os7 genes. (**A**) Phylogenetic tree showing the relationship of the predicted protein sequences of rice and Arabidopsis GH1 genes falling in At/Os7 and previously characterized Arabidopsis ABA-GE β-glucosidases (AtBG1 and AtBG2) and other At/Os phylogenetic clusters and two Arabidopsis specific clusters (At I and At II). Characterized rice proteins in other clusters, including those assayed for ABA-GE hydrolysis, and Arabidopsis SENSITIVE TO FREEZING2 (SFR2) and rice galactolipid transgalactosidase are labeled on unexpanded clusters. (**B**) Genomic map of the section on chromosome 4 containing Os4BGlu9-13.

**Figure 2 ijms-22-07593-f002:**
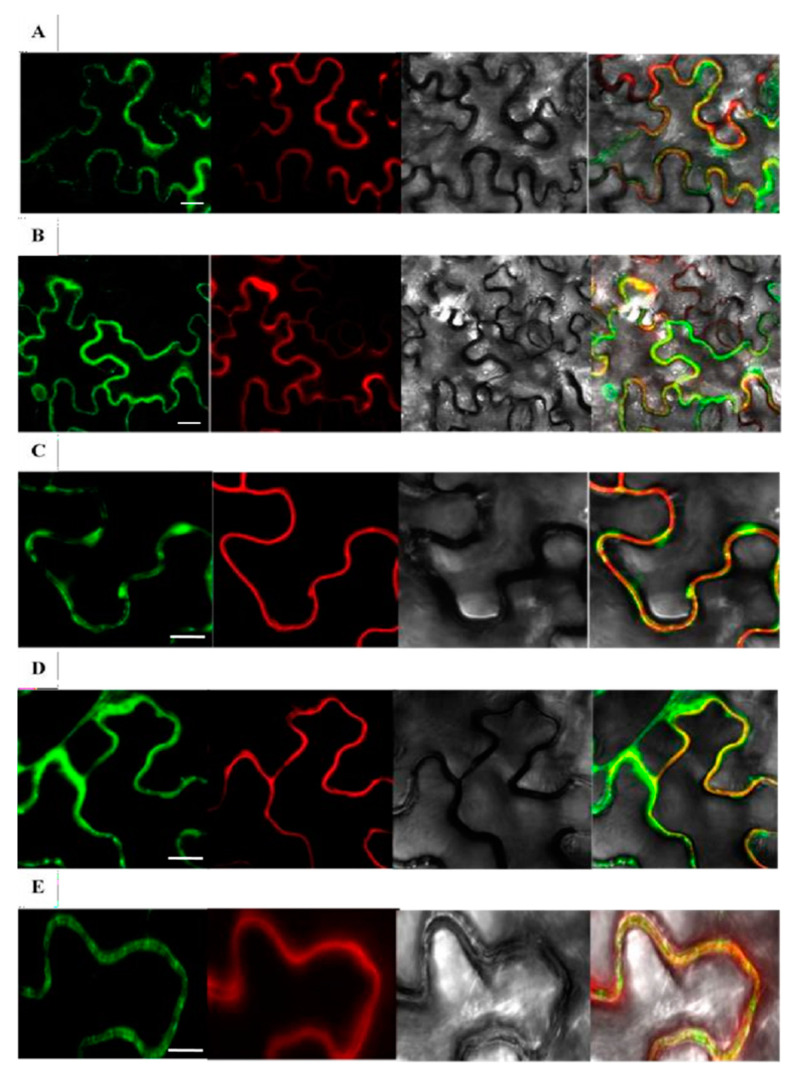
Subcellular localization of rice GH1 At/Os7 proteins in tobacco epithelial cells. Os4BGlu9-GFP, Os4BGlu10-GFP, Os4BGlu11-GFP, Os4BGlu2-GFP and Os4BGlu13-GFP co-infiltrated with RFP-KOR1 were transiently expressed in *N. benthamiana* leaves. The Arabidopsis Korrigan1 (AT5G49720) gene expression construct pCaMV35S:RFP-KOR1 was co-infiltrated into the leaves with pCaMV35S:Os4BGlu9-GFP (**A**), pCaMV35S:Os4BGlu10GFP (**B**), pCaMV35S:Os4BGlu11-GFP (**C**), pCaMV35S: Os4BGlu12- GFP (**D**) and pCaMV35S:Os4BGlu13-GFP (**E**). At 3-4 days after infiltration, the epidermal cells were plasmolyzed by infiltrating 0.8 M mannitol. Microscope images were captured at 10 min after the infiltration of leaves with mannitol by confocal scanning microscopy. For each gene, the images are from left to right: GFP fluorescence, RFP fluorescence, light microscopy, and merged images. Bars = 30 μm (**A**,**B**), 15 μm (**C**–**E**).

**Figure 3 ijms-22-07593-f003:**
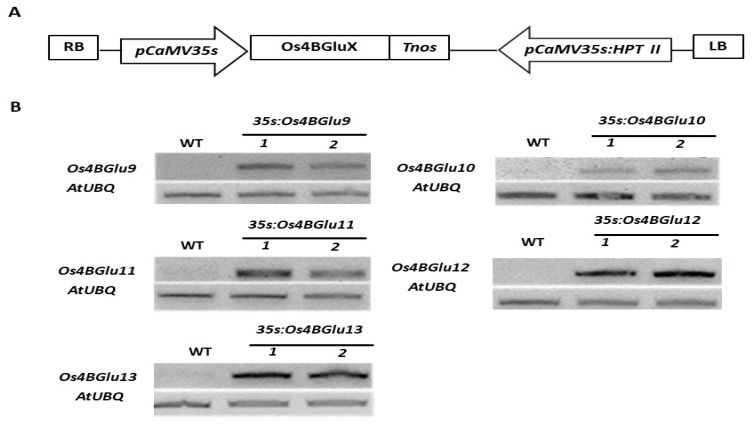
Heterologous expression of rice GH1 Cluster At/Os7 genes in Arabidopsis. (**A**) Schematic diagram of expression vectors in which Os4BGluX represents the cDNA for Os4BGlu9, Os4BGlu10, Os4BGlu11, Os4BGlu12 or Os4BGlu13. Cloned full-length cDNAs including partial sequence of 5′ and 3′ untranslated regions were inserted under the control of the pCaMV35S promoter. LB and RB, left and right borders of T-DNA; HPT II, hygromycin phosphotransferase II gene. (**B**) RT-PCR analysis of β-glucosidases gene expression in *Arabidopsis thaliana* wild type ecotype Col-0 to two independent lines of transgenic Arabidopsis expressing rice β-glucosidases Os4BGlu9, Os4BGlu10, Os4BGlu11, Os4BGlu12 and Os4BGlu13, respectively.

**Figure 4 ijms-22-07593-f004:**
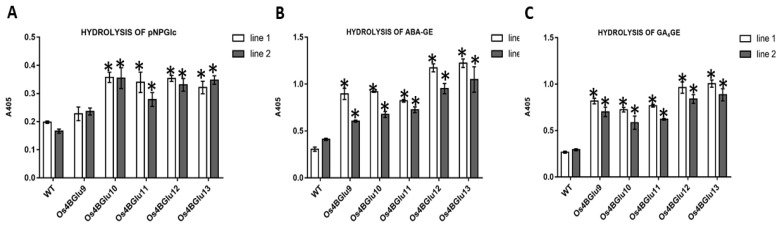
Hydrolysis of ABA-GE, GA_4_-GE and *p*NPGlc by plant extracts. Hydrolysis activity of extracts of wild type Col-0 Arabidopsis and transgenic Arabidopsis expressing rice β-glucosidase with *p*NPGlc (**A**), ABA-GE (**B**) and GA_4_-GE (**C**). Two independent lines of transgenic Arabidopsis expressing rice β-glucosidases Os4BGlu9-13 were grown under the same conditions. Three separate plant extracts were made for each line and extracts were incubated at 30 °C for 4 h in each substrate reaction. The asterisks (*) above the bars indicate significant differences from both control lines with a *p*-value < 0.05.

**Figure 5 ijms-22-07593-f005:**
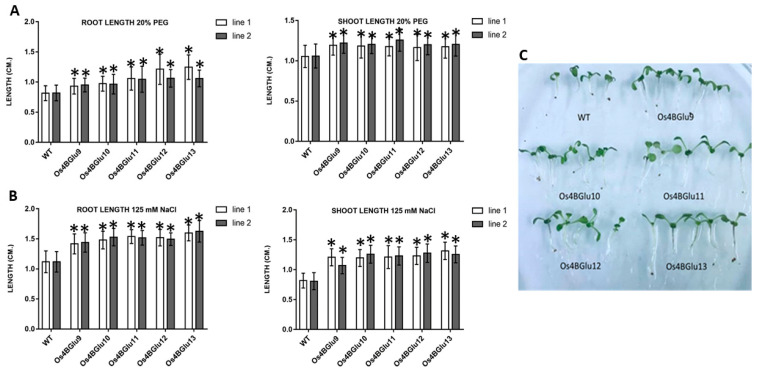
Effect of NaCl and osmotic stress on wild type and rice β-glucosidase-expressing Arabidopsis seedling roots and shoots. Control Arabidopsis plants and plants expressing Os4BGlu9-13 were grown on ½ MS plates for 7 d and transplanted to ½ MS containing 125 mM NaCl (**A**,**C**) or 20% PEG (**B**) for 5 d. To quantify growth response, root and shoot lengths were measured in three independent sets of 20 plants each for each line. White and black bars represent two lines of plants with heterologous expression of the same gene. Asterisks (*) above the bars indicate a significant difference from both control lines with *p* < 0.05.

**Table 1 ijms-22-07593-t001:** ABA-GE hydrolysis activity of rice and barley GH1 β-glucosidases produced by recombinant expression in *E. coli*. The activities were assayed by incubating 0.25 µg of enzyme with 1 mM ABA-GE in 50 mM buffer (sodium acetate, pH 5) at 30 °C for 30 min.

β-Glucosidase	Specific Activity (µM·mg^−1^·min^−1^)
Os4BGlu13	1.63 × 10^−3^ ± 0.005
Os4BGlu12	1.54 × 10^−3^ ± 0.027
Os1BGlu4	1.31 × 10^−3^ ± 0.019
Os3BGlu7	1.18 × 10^−3^ ± 0.040
Os4BGlu18	1.16 × 10^−3^ ± 0.026
Barley βII	0.57 × 10^−3^ ± 0.008
Os3BGlu6	0.39 × 10^−3^ ± 0.004
Os7BGlu26	0.11 × 10^−3^ ± 0.009
Os9BGlu31	0.06 × 10^−3^ ± 0.004

**Table 2 ijms-22-07593-t002:** Kinetic parameters of Os4BGlu12 and Os4BGlu13 for hydrolysis of ABA-GE.

β-Glucosidase	*K*_M_ (mM)	*k*_cat_ (s^−1^)	*k*_cat_/*K*_M_ (mM^−1^s^−1^)
Os4BGlu12	10.9 ± 0.9	7.50 ± 0.0002	0.689
Os4BGlu13	1.66 ± 0.09	20.6 ± 0.5	12.4

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
