# Peer review of "Action of Multiple Rice β-Glucosidases on Abscisic Acid Glucose Ester"

_ijms, 2021, doi:10.3390/ijms22147593_

Round 1

Reviewer 1 Report

Manuscript of Kongdin et al. elaborates on rice b-glucosidases acting on ABA-GE. The authors studied various aspects of the enzymes (their activity, subcellular localisation and involvement in salt/drought stress). The manuscript is moreless logically organised. However, I consider several aspects of the manuscript as insufficient as discussed below:

general limitations which need to be addressed

  1. The general impression from the paper is, that only positive results without any “negative control” are presented. That you played it to the safe side. By that I mean that you present data
    1. of enzymes which hydrolyse ABA-GE (Table 1), but none which would lack such activity. Did you really have such lucky hand when (randomly, I assume) you picked which enzymes will you test or did you just not bother to report the others? Of course, if you haven’t measured other enzymes, that’s fine and I would believe that all of the glucosidases were capable of hydrolysing this ester, but it would be good to mention it specifically. And if you
    2. even though you know that Os4BGlu12&13 hydrolyse at least SA and tuberonic acid glucosides, you have tested only ABA-GE for those two enzymes and only ABA-GE and GA4‑GE in the transgenic plants, not even trying other gibberellins or other glucosides (for example it would evoke to try other phytohormones). I understand, that you cannot test all glucosides present in plants, but if you know that these enzymes hydrolyse SA & TA glucosides, it would be only logical to add them to your analyses (plus other gibberellins, because they are not as much structurally different as ABA & GA4).
  2. I was truly disappointed that there is no determination of phytohormone levels in planta in the transgenic plants. I considered it such an obvious experiment, that I was surprised that it isn’t in the manuscript. That should be added to the manuscript. And not only ABAs and GAs, but the full spectrum to see whether other hormones were affected as well. Ideally, full metabolome could be measured to better grasp which compounds are hydrolysed by the enzymes.
  3. In the case of subcellular localisation examination, you need you use more controls (proteins located to other compartments) and maybe even other system to better determine the subcellular localisation of Os4BGlu12&13. Especially when you yourselves admit that some of the proteins may be localised elsewhere then is your marker. This is another example where you consider only the positive control and that is good enough for you.

less serious limitations (these are debatable, but in my opinion should be addressed)

  1. Explain in more detail the phylogenetics and the numbering of the genes/enzymes. What is the 4 in Os4BGlu9? Is that the chromosome? Are the last numbers unique for the whole genome or repetitive for each chromosome? E.g. is there Os3BGlu1, Os5BGlu2, Os1BGlu3, etc. or are there Os3BGlu1, Os3BGlu2, Os3BGlu3…, Os4BGlu1, Os4BGlu2, Os4BGlu3…, Os5BGlu1, Os5BGlu2…
  2. I highly recommend to switch rows and columns in Table 2, as I think that is more common layout
  3. you can simplify Figure 3A to a single diagram with “Os4BGluX” as they are all identical (as far as I could say) anyway
  4. you should put panel C in Figure 4 as panel A, as you are referring to it as the first one
  5. BTW it’s not overexpression if it’s gene from different organism. That is heterologous expression.
  6. You should not refer to second line (e.g. l. 201), but to second experiment as you have used different conditions for growth and obviously the results (in Figure 5) were more affected by conditions of the experiment, than by the line (as all the roots/shoots are either longer or shorter in experiment 1 than experiment 2, while if the line had higher effect, the change would be random). More appropriately, you should grow both lines at the same time at the same conditions. I see really no reason why should you grow them at various conditions, if you want to compare the results. Also, for this reason, I would recommend to express the results as % of WT plants. That should somewhat “normalise” the numbers.
  7. You should discuss in more detail the general function of ABA and GA and their involvement in response to abiotic stresses
  8. 258 – 259 – “production of secreted glucosidase might affect pools of ABA-GE in ER” – that is a little wild conclusion. The secreted proteins are often modified in ER and GA and thus unlikely to exert significant activity in the ER. Also, at least Os4BGlu9&11 appear to me to be in some “bulges” in the inward side of the PM (and thus would not be excreted – more controls are required, see above)
  9. The Methods section needs way more details.
    1. where did you get the Os9BGlu31 (Section 4.1)?
    2. Section 4.2 – did you express and purify all proteins in the same way? Because the papers are from different labs, so I doubt they all used the same methods.
    3. Section 4.4 – more details on the setup of microscope?
    4. Section 4.6 – please, consult this reference for more details that are required https://pubmed.ncbi.nlm.nih.gov/19246619/
  10. 338-340 – shouldn’t you provide rather ID from the KOME database? Also, you should provide citation for the KOME database.
  11. Section 4.7 – why did you grow the lines at different conditions? Why not grow them at the same time at the same conditions? Why did you grow your plant at three different temperatures (l. 386)?

some typing errors etc. (these are suggestions to improve the text, its readability and understandability, these are up to you, if you will follow them)

  1. The authors excessively split words at the end of lines. I found it quite disturbing and sometimes it was even wrong (e.g. l. 64, 165)
  2. on the other hand, sometimes you have hyphen at the end of line. For these, use the unbreakable hyphen (Ctrl+Shift+”-“ in MS Word; similarly unbreakable space is Ctrl+Shift+Space when you have e.g. units at the end of line etc.); e.g. l. 65, 89, heading of Table 1, l. 111, 180, 219, 228, 237, 251)
  3. I would recommend to slightly re‑phrase the sentence on lines 89-90 to “several glucosidases have been expressed in coli and screened.”.
  4. in Table 1, the heading says “Rice GH1 β‑Glucosidases”, but you have there also glucosidase from barley
  5. 115 – the plasma membrane
  6. 147 – in my opinion, there should be “…Os4BGlu12 or Os4BGlu13 cDNAs…”
  7. 178 – “glucosidase activity in the plant.”
  8. 192 – “their roots and shoots were all longer than control plants” – they were longer than roots and shoots of control plants
  9. 287 – in my opinion, there should be “production by Os9BGlu31 transglucosidase”
  10. use consistently italics in the abbreviation pNPGlc (e.g. lines 288, 290, 400, possibly elsewhere)
  11. 294 – you’re missing parenthesis at NMR and Ultra‑High should be with a dash
  12. 297 – 299 – “The hydrolysis activity … were expressed and purified…” 1) it’s grammatically wrong – sg. – pl. 2) you do not express/purify activity, but enzymes (or express genes). You may purify enzymes based on the activity, though.
  13. 306 – I think you can leave out the “as a blank” as that seems quite obvious
  14. 308-310 – I would recommend to re-write the sentence as follows: …were purified by IMAC followed by cleavage with enterokinase. The thioredoxin and His6 tags were removed by IMAC.
  15. 320 – I don’t think you need the sentence “belonging to cluster At/Os7 of GH1” in the Methods section. But if you want to keep it, you should probably write “to the cluster”.
  16. 373 – 125 mM NaCl or 20% PEG. (then you can simply also the following “Water potential was lowered by pouring approx.. 20 ml of PEG solution on top…)

In conclusion, overall, the manuscript is good, but there are some serious limitations of the manuscript. This is "just another" paper on glucosidases from your lab, so I would expect some synthesis of your knowledge (e.g. testing more compounds which you know that are substrates), but instead you play it safe. The rest are only (relatively) minor problems.

Author Response

Thank you for the comments that helped to revise our manuscript, please see the attachment. 

Reviewer 2 Report

The manuscript „ Action of multiple rice β-glucosidases on abscisic acid glucose ester” by Kongdin et al. concerns very interesting topic on β-glucosidases and its enzymatic abilities to hydrolize plant phytohormone glycoconjugates. In my opinion Introduction is clearly written and it includes all important information. I have also no major comments to Results and Discussion parts. The experiments were conducted with valid methodologies and data handling was appropriate. The findings are clearly reported in the text and a valid discussion section in perspective to the results and literature has been presented.

Although I generally find the article acceptable for the publication, there are a few things that in my opinion could be improved, namely:

In Table 1, unit of Specific Activity should be written as ‘µM·mg-1·min-1’.

In line 115 a letter is missing in the word ‘plasma’.

On Figure 2 there are missing bars, that are described under the Figure. In my opinion it would be better if the bars would be added to the Figure.

In Figure 4 and 5 descriptions, information about the meaning of an appendix ‘*’ should be added and it should be clearly indicate how the comparison was made. So it should be written if the comparison was made between Line 1 and 2 of each gene or if it was comparison between plants overexpressing Os4BGlu9-13 and control. It concerns also Supplementary Figure 1.

In Supplementary Figure 2, information about the Protein Marker 2 should be added. Who is producer of this marker?

Author Response

Thank you for your constructive comments. Please see the attachment.

Reviewer 3 Report

The manuscript “Action of multiple rice β-glucosidases on abscisic acid glucose ester” describes a functional analysis of β-glucosidase rice genes.

The study is of interest and in general well written, there is just one problem to fix.

Fig. 4 and 5: “Two independent lines of transgenic Arabidopsis over-expressing rice phytohormone β-glucosidases Os4BGlu9-13 were grown in two independent experiments (one line of each gene per experiment, white bar for the first experiment, black for the second). Plants were extracted and extracts were incubated at 30°C for 4 h in each substrate reaction.” It is not clear why the two transgenic independent lines were assessed in two independent experiments. One would expect one or two experiments each with all of the lines to be evaluated with two or three biological replications. Given the two independent experiments, it is difficult to compare results for the different lines.

A new experiment should be done with all of the lines with biological replications.

Round 2

Reviewer 1 Report

I have reviewed the revised version of the manuscript by Kongdin et al.  I have still a couple of comments, which follow with the original numbering, plus (unfortunately) a couple of new comments in the end:

    1. OK, but why did you pick this specific enzyme (Os7BGlu26) for the Discussion? Also, I would recommend such mention also in the Results section, e.g. on l. 94-97 you could write “We have tested rice Os4BGlu12… and barley βII, all of which showed activity, albeit low in some cases, as shown in Table 1.”
    2. I understand your difficulties. Maybe you could try company OlChemIm. They offer at least ABA-GE ( http://www.olchemim.cz/Products.aspx?idc=1&idp=5 ). Maybe if would you request the other compounds, they could be able to provide them as well.
  1. I understand, but there are in principle three possibilities how to overcome this:
    1. It is common to send samples for phytohormone analysis to laboratories which do such analyses on daily basis, namely
      1. Prof Sakakibara, Japan
      2. Prof Emery, Canada
      3. Dr Dobrev, the Czech Republic
      4. Dr Novák, the Czech Republic
    2. there may be commercially available ELISA kits for determination of your compounds of interest. Even if not, there is a way how to do it by yourself.
    3. you can always measure only the free phytohormones and observe, whether there is any change. Of course, you won’t get the whole picture, as there may be other levels of homeostasis, but better than nothing.

I would really love to see the results, but the Editor wrote me last time that you need this paper for graduation, so I’ll let it be for this time, but I hope to read in your next paper how  you overcame your obstacles.

  1. I meant more controls for other subcellular compartments (e.g. ER, if you think there may be significant amount of your enzymes). But I do like your way to distinguish the cell wall and plasma membrane localisation. So hopefully the next time… ;)
  2.  
  3.  
  4.  
  5.  
  6.  
  7.  
  8.  
  9. Good point with the E. coli. But I do not understand the last sentence (l. 304-306) – what builds up? Also, shouldn’t ABA-GE be in vacuole rather than in the ER?
  10.  
  11. I stand corrected. Previously, I got the impression that more papers were from other groups. But at least the citation 19 still appears to be from other group, isn’t it?
  12. Right, I have commented on it without thinking about it that much.
  13.  
  14.  
  15. Right, you reminded me of the great pain when I was working on a manuscript for MDPI :) I think I overcame it (where it was really bad) by turning it off for particular paragraph. (for example the pol-ymerase on l. 444 would deserve that)
  16. If you use unbreakable space/hyphen, it cannot be broken even by automatic changes. That’s why you use unbreakable space/hyphen. Try it on l. 266; 351; 372.
  17. 1) you use the same passive form as I suggested (I have only copied it from your text) 2) I meant to leave out “rice and barley” and “that”; the rest is the same as in your sentence. But this was only suggestion as I felt such sentence would be more pleasing, so it’s OK if you keep it as it is.
  18.  
  19.  
  20.  
  21.  
  22.  
  23.  
  24. l. 490
  25.  
  26.  
  27. In that case you could keep it as “The blank was incubated…”. If not, you have there twice “was” in the sentence.

  1. You’re using “(rice) phytohormone β-glucosidase” (e.g. l. 166, 188; 200; 203; 215; 473; 499). I would recommend not to use it for several reasons:
    • you’re not consistent. Sometimes you use it, other times you don’t
    • I assume that’s supposed to be in accordance to IUBMB recommendation that enzyme name includes substrate name, but in that case it should be phytohormone glucoside/glucose ester β-glucosidase
    • you could say something like phytohormone-GE-specific β-glucosidase, except you cannot, because you know the enzymes hydrolyse also other substrates
  2. l. 60 – shouldn’t there be “localizes to the vacuole”?
  3. l. 86 – belonging to the GH1 phylogenetic cluster
  4. l. 99 – which cluster? At/Os7?
  5. l. 168 – Figure 3A; l. 170 – Figure 3B
  6. l. 193 – …showed almost two-fold activity of…
  7. l. 196 – …Os4BGlu13 expressed in E. coli had high activity…
  8. l. 209 – were grown at the same conditions
  9. l. 220 – expressing rice β-glucosidases displayed
  10. The sentence on l. 221 doesn’t make much sense and it took me a while to realize that you’re comparing data among treatments (stressed vs. unstressed) rather than among genotypes (WT vs. OE), mainly because you refer to Supp. Fig. S2, which shows only data for unstressed plants. If you want to keep it this way, delete the “and” and use cf. Fig. 5 and Suppl. Fig. S2. Better yet, I would recommend something align the following: “Under unstressed conditions, the lines expressing rice β-glucosidases were indistinguishable from WT plants (Suppl. Fig. S2). Although salt and osmotic stress shortened the roots and shoots, the effect was significantly smaller for the transgenic plants.”

Moreover, in the paragraph, you first write about “roots&shoots”, then about “roots” followed by “shoots” only to finish with “roots&shoots” again. Which seems a little redundant.

  1. l. 264 – I would replace “apparent promiscuity” with “wide acceptability”, because enzyme promiscuity is IMHO property of an enzyme, not substrate
  2. l. 309-310 – I would leave out the whole “when it had been…less sensitive to ABA”
  3. l. 317 – 125 mM NaCl!
  4. l. 410 - …instructions. Thereafter, 20% sucrose was used as treatment on one side…
  5. l. 482 – https://www.ncbi.nlm.nih.gov/books/NBK995/#A261
  6. l. 487 – delete “and”

With the exception of points 31 and 40, they are only minor problems. I shouldn't need to see the manuscript again.

Reviewer 3 Report

I am satisfied with the revisions done.

Author Response

Thank you for the help improving our manuscript.